# Hypoxia in the Initiation and Progression of Neuroblastoma Tumours

**DOI:** 10.3390/ijms21010039

**Published:** 2019-12-19

**Authors:** Carlos Huertas-Castaño, María A. Gómez-Muñoz, Ricardo Pardal, Francisco M. Vega

**Affiliations:** 1Instituto de Biomedicina de Sevilla (IBiS), Hospital Universitario Virgen del Rocío/CSIC/Universidad de Sevilla, 41013 Seville, Spain; 2Departamento de Fisiología Médica y Biofísica, Universidad de Sevilla, 41009 Seville, Spain; 3Departamento de Biología Celular, Universidad de Sevilla, 41012 Seville, Spain

**Keywords:** neuroblastoma, hypoxia, neural crest cells, cancer stem cells, paediatric oncology

## Abstract

Neuroblastoma is the most frequent extracranial solid tumour in children, causing 10% of all paediatric oncology deaths. It arises in the embryonic neural crest due to an uncontrolled behaviour of sympathetic nervous system progenitors, giving rise to heterogeneous tumours. Low local or systemic tissue oxygen concentration has emerged as a cellular stimulus with important consequences for tumour initiation, evolution and progression. In neuroblastoma, several evidences point towards a role of hypoxia in tumour initiation during development, tumour cell differentiation, survival and metastatic spreading. However, the heterogeneous nature of the disease, its developmental origin and the lack of suitable experimental models have complicated a clear understanding of the effect of hypoxia in neuroblastoma tumour progression and the molecular mechanisms implicated. In this review, we have compiled available evidences to try to shed light onto this important field. In particular, we explore the effect of hypoxia in neuroblastoma cell transformation and differentiation. We also discuss the experimental models available and the emerging alternatives to study this problem, and we present hypoxia-related therapeutic avenues being explored in the field.

## 1. Introduction

Neuroblastoma (NB) is a paediatric solid tumour arising from the neural crest (NC) during sympathetic nervous system (SNS) development [1]. The NC is a migratory and multipotent embryonic cell population that gives rise, among other tissues, to the SNS [2]. Despite its low frequency, NB is the most common malignant extracranial solid tumour in children, and it is the cause of 10% of total child cancer deaths [3]. This cancer is characterised by great heterogeneity, including some very aggressive metastatic tumours which present poor response against conventional therapies and frequent relapses [4].

Local or systemic low oxygen exposure (hypoxia) has been linked to NC development, including neural crest cells (NCCs) migration and differentiation. Hypoxic conditions in the intratumoural microenvironment lead to the deregulation of multiple cellular processes, including cell proliferation, cell survival and cell invasion, generally leading to tumour aggressiveness and resistance to therapy. Despite a growing number of evidences, there is still little consensus related to the effect of hypoxia in NB initiation and progression. In this review, we summarise the main findings regarding hypoxia and the origin, progression and malignisation of NB. We also make an overview of the tools available for the study of this phenomenon and provide insights into possible therapeutic approaches under development.

## 2. Approaches to the Study of Hypoxia in Neuroblastoma

NB is a heterogeneous disease, with high intertumoural and intratumour heterogeneity, including genetic and functional diversity of cells, as well as a diverse tumour microenvironment [5]. The study of the effects of hypoxic conditions on tumour initiation and tumour progression should take into account this emerging reality. The experimental models available for the study of hypoxia in tumours, and in particular in NB, greatly affect the results obtained. Moreover, differential set ups should be used to study its influence in tumour initiation, progression or metastasis. Currently available NB models comprise genetically engineered mouse models (GEMMs), patient-derived models (patient-derived xenografts, PDXs) and cell lines derived from them or immortalised in culture from patients’ samples. In addition, developmental models such as chick embryo and zebra fish have proven useful to study the generation of NB from the neural crest.

Usually, two main methods are used for controlling O_2_ concentration exposition in cell culture or in vivo experiments: environmentally controlled chambers for cells or experimental animals, or a biochemically induced state of pseudohypoxia, mostly by using cobalt chloride [6]. The study of hypoxia in vitro on cultured cells offers ample opportunities of analysis and manipulation in a stable environment but reflects the heterogeneity and physiology of a patient tumour poorly, and ignores nutrient deprivation, tumour–stromal–extracellular matrix (ECM) interactions, and angiogenesis. Most knowledge obtained so far has been derived from experiments performed on cultured cells.

NB GEMMs are a powerful tool to study tumour heterogeneity and to follow tumour evolution. In addition, spontaneous tumour formation can be tracked. Nonetheless, tumour formation and evolution in GEMMs is inherently associated with genomic aberrations introduced and with their induced specific secondary genetic events. The best-known NB GEMM used to date is the tyrosine hydroxylase *TH-MYCN* model, a transgenic mouse that expresses *MYCN* (neuroblastoma-derived-*v-myc* avian myelocytomatosis viral related oncogene) under the control of the rat *TH* promoter, activated during development, and its derived versions [7,8]. This model has proven useful for stepwise mechanistic analysis of the interaction between tumour and its vascular compartment and for the study of angiogenesis, which makes it a promising model to study hypoxia and tumour progression in NB [9]. Although the *TH-MYCN* model has been shown to recapitulate the features of MYCN amplified human tumours, it presents some drawbacks, such as a variable tumour incidence depending on background and a low incidence of metastasis. These characteristics impose difficulties for some hypoxia-related studies. New spontaneous NB tumour models are needed. For instance, a *TH-MYCN*/*Casp8*^−^^/^^−^ mouse model has shown a higher incidence of bone marrow metastasis and more new models are appearing with promising results [10].

The study of hypoxic events in NB tumour initiation presents special challenges. Very little is known about these early hypoxic events. Moreover, the lack of recurrent somatic mutations complicates the generation of meaningful models. It would be desirable to enhance the *TH-MYCN* or other transgenic mouse models to be able to follow malignant NC cells during their transformation. Chick embryo or zebrafish models of NC development are useful for early events but do not offer a tumour progression close enough to humans. Recently, new primary NB cell lines have been developed by conditional reprogramming of cells cultured from *TH-MYCN* mice tumours recapitulating the heterogeneous phenotypes present in tumour populations [11]. This could prove a useful tool to study tumour initiation in the interphase between in vitro and in vivo.

PDXs are likely the most encouraging model for studying the biology of human NB, including their response to hypoxia. Although the use of PDXs for this type of studies has been limited in scope and number, patient-derived orthotopic xenografts (PDOXs) retain geno- and phenotypes of patient tumours and metastatic patterns, recapitulating most of the microenvironmental hallmarks of aggressive patient tumours, despite lacking a complete host immune system [12]. Therefore, PDOXs could be an optimal model for studying hypoxia during NB tumour engraftment and progression. In addition, given the links between hypoxia and NB cell differentiation, it is useful to work in a tumour model where the original tumour heterogeneity can be maintained and is not constrained by the particular genetic driver of GEMM models [13].

Due to the cost and infrastructure requirements for the use of PDOXs, there is an interest in generating cell-based culture systems derived from patient samples that can mimic most patient tumour characteristics without compromising versatility. The generation and propagation of tumour spheroids and the establishment of NB organotypic cultures are suitable alternatives under development [14,15]. While in vivo studies using PDXs are not compatible with high throughput analysis, the use of 3D cultures may fill the gap between monolayer cell culture assays and model animal experiments, as seen in Figure 1. Various devices for the specific study of hypoxia in cells, tissues and their microenvironment have been developed [16]. Hopefully, the combination of all these complementary models will give us a clearer idea of the mechanisms and processes involved in NB progression affected by oxygen concentration.

## 3. Hypoxia and the Origin of Neuroblastoma

NCCs are formed at the border of the neural plate, where they undergo epithelial–mesenchymal transition (EMT) processes, leaving the neuroepithelium in the delamination phase to migrate all over the embryo. Once they reach their destination, NCCs differentiate into many different cell types: pigment cells, sympathetic neurons, glial derivatives, smooth muscle cells, odontoblasts, or adipocytes. Most NBs are found in the adrenal medulla or sympathetic ganglia, organs with a sympathoadrenal lineage aetiology derived from the NC. Thus, NCCs have been described to be the precursors of this malignancy. Failures of NCC migration or differentiation mechanisms are thought to result in the development of NBs. Different genetic events have been described to be related with NB initiation; for example, chromosomal aberrations such as *MYCN* amplification, chromosome 11q region loss or 17q gain. More recent research points to somatic mutations in genes such as anaplastic lymphoma kinase (*ALK*), phosphatase and tension homolog (*PTEN*) or alpha thalassemia/mental retardation syndrome X-linked (*ATRX*), among others [2,17].

Deregulation of blood oxygen levels during embryonic development can interfere with neural crest-derived tissue maturation. The relevance of hypoxic events during normal embryonic and foetal development affecting placentation, angiogenesis and haematopoiesis has been widely described [18,19]. Hypoxia regulates the amount of neural crest migrating cells in a chick embryo model [20]. The expression in NC migrating cells of genes involved in NCC delamination, such as *SNAIL2*, *SOX10* or C-X-C chemokine receptor type 4 (*CXCR4*), was higher under hypoxic conditions, stimulating the migration of NCC. Hypoxia seems to induce proliferation in rat neural crest stem cells [21]. Both phenomena were dependent on hypoxia inducible factor 1 alpha (HIF-1α), a transcription factor typically stabilised and activated under hypoxic conditions. HIF-1α expression is needed for normal embryonic development and HIF-1α knock-out mice present severe neural tube defects [22]. The activation of HIF-1α by the platelet-derived growth factor 1 alpha (*PDGF1α*) and phosphatidyl inositol kinase 3/phospho alpha serine-threonine protein kinase (PI3K/AKT) signalling pathway is also required for NCC EMT and migration, as shown in *Xenopus* and zebrafish embryo models [23].

While HIF-1α stabilisation is normally involved in the response to hypoxia by the majority of tissues, HIF-2α activation and stabilisation has been linked to the response to chronic hypoxia by NCC, including NB cells [24]. HIF-2α knockout models present a variable developmental phenotype, including vascular disorganisation, catecholamine levels reduction, bradycardia and midgestational death. An important role for HIF-2α in SNS development has been described, participating in the activation of aromatic L-amino acid decarboxylase (DDC) and dopamine beta-hydroxylase (DBH) transcription in rat sympathoadrenal progenitors, resulting in a mature sympathoadrenal phenotype. Failures in the regulation of HIF-2α result in the increment of sympathetic cells due to a reduced apoptotic rate, giving rise to dysfunctional adrenal medulla and carotid body tissues [25].

There is preliminary evidence pointing towards a role for low oxygen levels exposition in NB initiation during development. A clinical study about prenatal and perinatal events and neonatal risk factors in NB showed that maternal anaemia, neonatal respiratory distress or neonatal haemolytic disease could increase the risk of developing NB [18]. HIF-2α expression has been found in foetal paraganglia and in a progenitor cell population in NB samples, correlating with vascular endothelial growth factor (VEGF) expression [24]. Therefore, it has been suggested that HIF-2α deregulation could play a role in the malignant transformation of sympathoadrenal progenitors, giving rise to NB tumours. Hypoxia has also been related to other events connecting NC development and NB. For instance, the frizzled class receptor 6 (frizzled-6) of the Wnt pathway, responsible for NC induction, is also highly expressed in hypoxic areas on NB tumours [26], further linking NC development with NB and hypoxic conditions. 

Tumour hypoxia can also lead to genomic instability. Hypoxia has been widely described to be involved in deregulation of DNA repair and damage response genes expression, reactive oxygen species (ROS)-dependent acquired mutations, cell cycle arrest, DNA replication alterations or chromosomal and microsatellite aberrations [27,28]. A recent study across 19 tumour types revealed a hypoxia signature associated with high genomic instability [29]. These facts raise the question whether hypoxic events during embryonic development could have an active role in the generation of genomic instability events associated to NB initiation, such as the incidence of a massive chromosomal rearrangement, known as chromothripsis [30]. For instance, hypoxia could activate the alternative neurotrophin receptor tropomyosin-related kinase A III (*TrkAIII*) splicing variant that promotes genetic instability by its interaction with the centrosome [31].

Despite the prominent role of hypoxia during NC development and emerging evidence suggesting a role for hypoxia in NB initiation, the knowledge about this topic is still very limited and highly speculative, due to the limited approaches to properly address NB initiation events.

## 4. Neuroblastoma Differentiation and Stemness in Hypoxic Contexts

The correlation between NB tumour differentiation, prognosis and clinical outcome is well established, with a worse outcome attributed to tumours with undifferentiated histology [32]. Furthermore, NB cellular heterogeneity, and in particular the presence of an undifferentiated neural crest stem-like cell population, has been proposed to have a fundamental effect on tumour progression and response to therapy [33,34]. Putative markers for cancer stem cells (CSCs) in NB have also been described, although there is not a definitive conclusion on this matter [35]. Local oxygen tissue concentration exerts determinant relevant roles in tumour microenvironment, affecting cell differentiation and stemness in turn.

There has been some controversy regarding the differentiation potential of hypoxic conditions in NB tumours. Early studies suggested that hypoxia promotes aggressive tumour behaviour in NB, showing a neuronal-to-neuroendocrine lineage shift in cells close to non vascularised necrotic areas [36]. Works based on the morphological and molecular analysis of 116 clinical NB tumours, and supported by hypoxia-induced phenotypes observed in vitro and in vivo, showed that hypoxic NB cells switch from primitive neuronal to a chromaffin cell type, which was reversible upon reoxygenation [36]. These regions were characterised by the expression of high levels of the insulin-like growth factor 2 gene (*IGF2*) and low levels of the growth-associated protein-43 gene (*GAP-43*), expressed in chromaffin and neuronal cells, respectively, in the foetal nervous system. Same results were obtained assessing the expression of neuroendocrine secretory protein 55 (NESP55), a sensitive early human sympathetic nervous system chromaffin marker, concluding that 0.5–1% O_2_ triggers hypoxia-mediated chromaffin metaplasia across the clinical spectrum of neuroblastic tumours [37].

By contrast, results from other groups suggest the induction of immature, stem-cell-like features in response to hypoxia, and associated with poor clinical outcome. Authors identify the expression of markers for immature neural crest cells, such as inhibitor of DNA binding 2 (*ID2*), and the downregulation of neuroendocrine enzymes, such as chromogranin A, in NB cells after exposition to hypoxia [38,39]. A global gene expression analysis showed that hypoxia provoked a general adaptive response in NB cells, confirming loss of the neuronal phenotype in favour of a gain in stem cell genes [40]. An upregulation of proneuronal lineage specifying transcription factors inhibitors (*ID2* and *HES-1*) and downregulation of proneuronal lineage specifying transcription factors was found. However, they also described the expression of genes involved in the physiological response to low oxygen, such as TH or DDC, indicating that the response might be cell-type-dependent.

Hypoxia-induced dedifferentiation of tumour cells, as a mechanism behind cellular heterogeneity and aggressiveness in solid tumours, has been subsequently reviewed and supported by multiple works [41]. HIF-1α and HIF-2α have been observed to be differentially recruited to target genes in NB in response to hypoxia, with HIF-2α preferentially promoting an aggressive phenotype [24]. HIF-2α regulates the *OCT4* gene influencing stem cell function, embryonic development, and tumour growth [42]. A perivascular niche for brain tumour stem cells has been described [43] and the brain is one of the main metastatic sites for aggressive NB. Therefore, the existence of a perivascular niche for NB stem cells has been explored. High levels of HIF-2α delineate an immature neural-crest-like NB cell subpopulation, characterised by the expression of NOTCH1, HES-1, c-Kit, vimentin and dHAND, and located in a perivascular niche [44]. This hypoxia-inducible factor has been associated with increased invasion, tumourigenicity and stemness [45]. A meaningful function for HIF-2α in maintaining an undifferentiated state in neural-crest-like human NB tumour-initiating cells has been proposed [46]. It seems clear that HIF-2α protein levels in NB, as well as an immature phenotype, correlate with an unfavourable outcome, although potential hypoxia-independent functions of HIF-2α, related to aggressiveness, have also been suggested. Proteomic analysis in NB cells shows that increased hypoxia-independent HIF-2α expression correlates with an increment of fatty acid storage and a shift in global splicing regulations that has been previously reported in high-risk tumours [47,48].

Global gene expression analysis has also been used to investigate the relationship between hypoxia and NB progression, particularly analysing the role of HIF-2α. In support of the induction by hypoxia of immature stem-cell-like features contributing to a malignant phenotype in NB, Applebaum et al. integrated diverse transcriptomics datasets from NB tumours and cell lines, identifying a hypoxia-upregulated signature correlated with poor patient outcome in three independent cohorts [49]. The signature included the expression of *LCO4A1*, *ENO1*, *HK2*, *PGK1*, *MTFP1*, *HILPDA*, *VKORC1*, *TPI1*, and *HIST1H1C.* They experimentally validated the findings demonstrating upregulation in response to hypoxia in four NB cell lines. Other hypoxia-related gene signatures have been used to predict patient outcome, with poor outcomes for patients with hypoxic tumours, supporting the potential of using hypoxia-related pathways as targets for NB treatment [50,51].

On the other hand, a recent study suggests that HIF-2α expression correlates with features of low-risk NB and with adrenal chromaffin cell differentiation [52]. In this work, HIF-2α high expression correlates with significantly increased overall survival and with genes associated with differentiation and better patient outcome. In addition, treatment with HIF-2α inhibitors did not block in vitro NB cell proliferation or xenograft tumour growth. Single cell sequencing data from the developing mouse sympathoadrenal lineage has allowed the elucidation of the cellular compartments presenting high levels of *Hif-2α* [53]. This analysis suggests that HIF-2α expression is generally high in chromaffin differentiated cells and in low risk, MYCN nonamplified tumours, precluding its consideration as an oncogene in NB.

This controversy in the field might be clarified in light of emerging studies highlighting the cellular heterogeneity present in NB, and demonstrating the presence of different transcriptomic circuits in phenotypically different NB cell populations [33,54]. In general, and despite the contradictory results involving HIF-2α, hypoxia seems to evoke a more aggressive NB phenotype, favouring the undifferentiated immature state of stem-like cells and correlating with poor prognosis and worse patient outcome. However, hypoxia might also promote a switch to mature chromaffin cells in a different, more determined, cell population. Given the transcendence of undifferentiated cancerous cells in tumour evolution, it would be essential to study if their persistence could be favoured by hypoxic environments during cancer initiation and progression.

## 5. Hypoxia and NB Cell Survival

Hypoxic conditions in intratumoural microenvironments lead to the deregulation of multiple cellular processes. Experimental and clinical data associate cellular response to hypoxia with tumour aggressiveness and resistance to therapy in many cancer types, which is linked with higher proliferation rates and apoptosis bypass [55].

A global gene expression analysis performed in NB-derived cells under normoxic/hypoxic conditions found upregulation by low oxygen of cell survival and growth related genes such as *VEGF*, *NEUROPILIN 1*, *ADRENOMEDULLIN* and *IGF2*, among others [39]. IGF-1 and IGF-2 upregulation, together with the expression of VEGF, was proposed as a possible activation of an autocrine signalling loop that would allow cells to overcome low oxygen concentration in the hypoxic tumour microenvironment. A transient cell cycle arrest, but no increase in cell death, was described after hypoxic exposition in SK-N-BE NB cells. Cell cycle arrest has been described in renal clear cell carcinoma (RCC) and colon cancer cell lines in relation to HIF1-α stabilisation. On the contrary, HIF-2α stabilisation was described to drive cell cycle progression in RCC cells in a c-myc dependent manner [56]. However, not enough data is available regarding hypoxia and proliferation in NB. A relationship between MYCN amplification and HIF-1α expression driving high proliferation has been suggested, although the connection with hypoxia has not been established [57].

More information exists on hypoxia-related apoptosis resistance. Hypoxia exposition for more than 12 hours has been described to induce apoptosis in PC12 cells, but HIF-1α activation would promote antiapoptotic pathways via caspase 3 and Ras homolog family member A (RhoA) activation [58]. No change in Ras homolog family member (RhoB) expression after hypoxia was found in this study, which is in contradiction with other published data [59]. MYCN amplification has been described to correlate with HIF1-α stabilisation, exerting a possible antiapoptotic effect through Survivin upregulation [60]. VEGF expression in response to hypoxia has been described to trigger apoptosis resistance in NB cells by the upregulation of B cell leukaemia 2 (Bcl-2) expression and extracellular-signal-regulated kinase (ERK1/2) phosphorylation [61,62,63]. ERK1/2 was found also to phosphorylate HIF-1α, keeping VEGF expression in an autocrine-loop.

This resistance to apoptosis might be partially responsible for chemoresistance associated with hypoxic conditions in cancer cells [64]. Cellular adaptation to hypoxia provokes HIF-1α-mediated resistance to chemotherapeutic agents [65,66]. NB cells adapt to hypoxia by HIF1-α-dependent and HIF1-α-independent responses. HIF1-α driven transcriptional response in hypoxia is a consequence of the epigenetic control of low oxygen levels at DNA methylation status in intragenic and intergenic regions [67,68,69]. Therapy resistance in cancer cells could be triggered by many different cellular events, such as emergence of DNA instability, drug metabolism, drug efflux, cell cycle arrest or apoptosis inactivation. Hypoxia has been described to drive misregulation of DNA repair and damage response genes, such as Ataxia Telangiectasia mutated (*ATM*) or *ATRX,* in cancer cells, and has been shown to increase rates of mutations due to intracellular ROS production [61]. Cell cycle hypoxia-related deregulation could also lead to aberrant proliferation of damaged cells, making them susceptible to acquired drug-resistance. Metallothioneins, proteins related to drug resistance, have been found to be induced under hypoxic conditions in NB cell lines [40]. p53 is barely mutated in NB, but it is believed that p53 response inactivation could be associated with relapses and multidrug resistance [70]. Furthermore, both p53 and HIF1-α are degraded/stabilised, respectively, by murine double minute 2 (MDM2), and it has been proposed that these two factors together could be interplaying in some way [71]. p53 has been described to be activated under hypoxia, thus increasing apoptosis. However, loss of sensitivity to p53-mediated apoptosis could occur further downstream in the signalling cascade.

## 6. Hypoxia and Metastasis in Neuroblastoma

In NB, metastatic disease is detected in approximately 50% of patients at diagnosis, frequently in the regional lymph nodes, bone marrow and bone [72]. Hypoxia is linked to metastasis, but how it affects metastatic progression is not clear due to limited consensus in the literature. Sustained hypoxia in a growing tumour seems to be associated with increased invasive capacity, perifocal tumour cell spreading and regional and distant dissemination, resulting in a clinically aggressive phenotype [55,73]. Hypoxia can also decrease motility speed, while increasing invasiveness and directionality, and even induce a dormancy-like program [74,75]. The molecular mechanisms responsible for these effects are not completely understood.

On NB, the number of published works dealing with this topic is low. In NB cells, hypoxia promotes the expression of *TRIO* and *SERPINB9* [24], genes thought to be involved in metastasis in other cancers [76,77,78,79]. A prometastatic gene expression program activated by hypoxia in NB cells has been described [80]. In vitro experiments using chemically induced hypoxia showed the downregulation of genes involved in maintenance of cell–cell and cell–matrix interactions, such as metalloprotease-4 inhibitor (*TIMP4*), E-cadherin (*CDH1*), catenin (*CTNNB1*) and fibronectin-1 (*FN1*). In contrast, metastatic behaviour genes, including integrin α7β1 (*ITGA7*), hepatocyte growth factor receptor (*HGFR*), transforming growth factor-β1 (*TGFβ1*), *VEGF* and interleukin-1β (*IL1B*) were upregulated under the same conditions. Hypoxia seems to contribute in NB cells to invasiveness through HIF-1α via sonic hedgehog signalling (SHH) [81]. Blockage of SHH signalling and the glioma-associated oncogene (*GLI*) transcription factor could be an effective way to target high-risk metastatic NB [82]. In vitro experiments show that HIF-1α enhances NB cell migration under hypoxic conditions maybe through a mechanism involving matrix metalloproteinases (MMPs) [81].

The effect of hypoxia in metastasis can be better highlighted by in vivo experiments, when a real microenvironment is in place. Experiments performed on chick embryos showed that hypoxia provokes NB cell invasion and metastasis. Furthermore, invasion by nonmetastatic cells can be triggered by close contact with metastatic cells [83]. Hypoxia preconditioning was sufficient to activate the long-lasting transcription of several prometastatic genes, including *MMP2*, *TWIST1*, *SNAIL1* and *CDH1*. HIF activity was indispensable for the acquisition of the metastatic phenotype.

Most studies pointing towards a possible involvement of hypoxia regulation in NB metastasis are indirect and come from the observation of HIF-dependent effects. For example, overexpression of MDM2 leads to increased HIF-1α transcriptional activity, while in vitro inhibition provokes decreased transcription of downstream targets [84]. Under hypoxic conditions, Nutlin-3a, a specific MDM2 inhibitor, suppresses HIF-1α expression, leading to p53-independent inhibition of VEGF [85]. Nutlin-3a also significantly reduced the incidence of NB metastasis in vivo. Isatin, an inhibitor of monoamine oxidase (MAO), which in turn stabilises HIF-1α, inhibits cell invasion and metastasis in NB through a mechanism involving the SDF1/CXCR4 axis [86]. It is, however, unclear whether this mechanism is regulated in any way by hypoxia. In NB, microRNA-145 (miR-145) induces HIF-2α repression, thus promoting downregulation of downstream cyclin D1, MMP14 and VEGF [87]. miR-145 suppresses NB cell growth, invasion and angiogenesis in vitro and inhibits metastasis in vivo in a HIF-2α dependent manner.

So far, despite the accepted view of hypoxic conditions affecting metastasis, including in NB, the molecular mechanisms surrounding these effects have been poorly elucidated. In this sense, limitations of the standard assays used to evaluate hypoxia in cancer and metastasis, mostly based on in vitro approaches, are a major handicap.

## 7. Hypoxia-Related Therapies

As discussed in this review, hypoxic events could have a fundamental impact on NB initiation, progression and metastasis. The finding of new targets and development of new therapeutic approaches is a main concern in the field. The lack of suitable, standardised in vivo models and the little knowledge about the mechanisms involved hinders more rapid progress. One of the best described effects of hypoxia in tumours is the triggering of an aggressive angiogenic response. Blocking tumour angiogenesis has been one of the most studied therapeutic approaches in recent years [88]. In NB, HIF-2α positive cells, expressing immature neural crest cell markers, were found in a perivascular niche; therefore, targeting them with antiangiogenic therapy arose as a new treatment strategy, with some success in vivo [89,90]. Many studies have focused on VEGF targeting as one of the most relevant approximations. Indeed, suppressive expression of VEGF has been described to increase the effectiveness of cytotoxic Irinotecan treatment in NB animal models [91]. Targeting of VEGF with the receptor-tyrosine kinase (RTK) inhibitor Sunitinib has been shown to be effective in NB metastatic cell treatment, but no change in resistance to this therapy has been found in clinic [92]. Sunitinib has been combined in another study with Evofosfamide, a cytotoxic prodrug activated under hypoxia, showing enhanced effectivity in tumour growth suppression [93]. However, antiangiogenic therapies have been shown to trigger metastasis in some models, precluding the general use of this approach at this point [90].

Some authors have suggested HIF targeting as a viable strategy to overcome resistance to therapy. HIF-1α expression decreases after Sunitinib treatment and HIF-1α activation has been described in cells resistant to different therapies [94]. In NB, HIF-2α has been more directly related to poor prognosis. HIF-2α inhibition leads to VEGF reduction [95]. The introduction of HIF inhibitors into differentiation therapy protocols might offer therapeutic advantages for relapse prevention. Indeed, the combination of all-trans retinoic acid (ATRA) with HIF-1α or HIF-2α silencing leads to an acquired glial cell phenotype and enhanced expression of glial cell differentiation markers. HIF-1α or HIF-2α silencing might also promote cell senescence independent of ATRA treatment [96]. However, some reports link HIF-2α inhibition to a decreased tumour response to treatment [97]. Thus, the targeting of HIF-2α in NB tumours is still a questionable strategy. More detailed studies related to the targeting of hypoxic zones in the tumour and hypoxia cellular response mechanisms are needed to advance in this direction.

## 8. Future Perspectives

Understanding the molecular mechanisms involved in NB initiation and progression is a major concern in order to improve the prognosis and treatment of this devastating disease. The data available supports the idea of hypoxia having an impact in NB development, NB cell fate and metastasis activation, among other processes, as seen in Figure 2 and Table 1. However, there is still great controversy about the role of hypoxia in NB cell differentiation, cell survival balance or NCCs malignant transformation. Furthermore, therapeutic approaches targeting hypoxia-related processes tend to fail in the clinic due to emerging resistance mechanisms.

Deeper knowledge about the molecular mechanisms underlying hypoxia response in NB cells is needed to fully understand the biology of the disease and provide new therapeutic opportunities. For that, the development of new laboratory models and tools, including the use of in vivo models of hypoxia and 3D in vitro cellular assays, would be necessary in order to clarify the contradictory results obtained in divergent cellular models. These approaches should consider the important relevance of intratumour cellular heterogeneity to delineate population-specific responses to hypoxia. In particular, the elucidation of the effects of hypoxia exposition in the phenotype and character of the undifferentiated NB neural crest stem-like cell population would help clarify the role of hypoxia, both in NB initiation and in malignant progression.

## Figures and Tables

**Figure 1 ijms-21-00039-f001:**
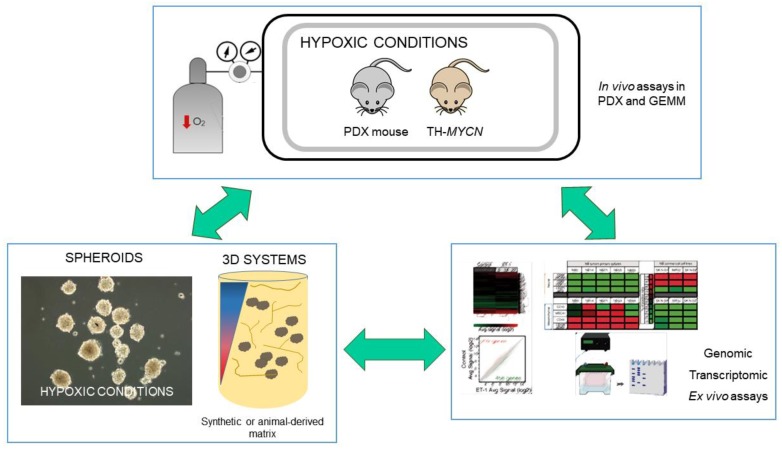
Experimental approaches for the study of hypoxia in neuroblastoma. Connecting molecular characterisation with information directly obtained from in vivo assays in hypoxia chambers and/or in vitro cultured spheroids would allow the generation of relevant evidence-driven hypothesis. 3D systems can be used for ex vivo or in vitro testing. Decreased oxygen concentration is symbolized in the illustration as red arrow pointing down or color gradient blue to red. Green arrows denote possible workflows.

**Figure 2 ijms-21-00039-f002:**
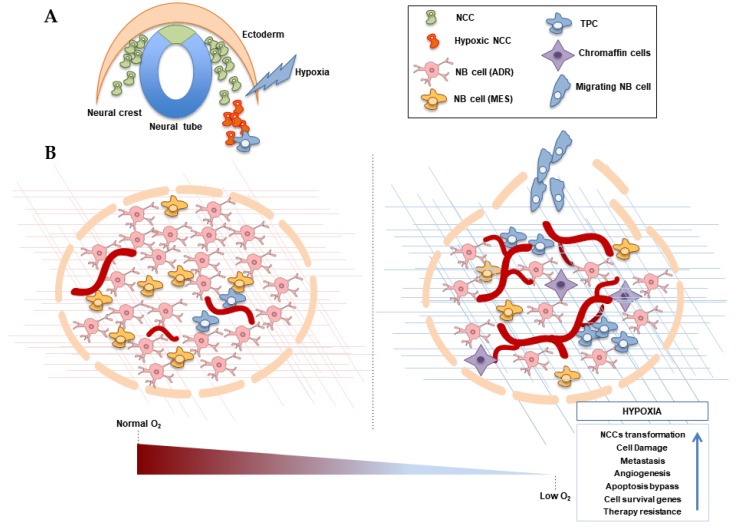
Effects of hypoxia in NB initiation and progression. (**A**) Neural crest (NC) migrating cells can undergo malignant transformation due to hypoxia exposition or disregulation, driving tumourigenesis. (**B**) A neuroblastoma (NB) tumour is comprised of phenotypically divergent cells. Hypoxia might favour malignant progression in NB tumours, characterised by the activation of angiogenesis, alteration of cellular differentiation, tumour cell survival and resistance and an increase of metastasis. The extracellular matrix is also affected by hypoxia. NB: neuroblastoma; NCC: neural crest cell; ADR: adrenergic cells, MES: mesenchymal undifferentiated cells; TPC: Undifferentiated tumour-propagating cells. Blue arrow in box denote an increase.

**Table 1 ijms-21-00039-t001:** Hypoxia-mediated gene-expression regulation in neuroblastoma.

Cellular Processes	Genes		References
**Initiation**	↑ *HIF-1α, HIF-2α, SNAIL-2, SOX10, CXCR4, PDGF1-α, FZD-6, TRKAIII*		[21,22,23,25,26,31]
**Differentiation**	↑ *IGF-2, NESP55, ID2, HES-1, TH, DDC, HIF-2α, OCT4, NOTCH1, c-Kit, Vimentin, dHAND*	↓ *GAP-43, CHGA*	[37,38,39,42,44]
**Cell survival**	↑ *HIF-1α, HIF-2α, VEGF, NEUROPILIN 1, ADRENOMEDULLIN, IGF-1, IGF-2, RHOA, BCL-2, Survivin, Methallothioneins, p53, MDM2*		[39,54,56,58,59,60,61,68]
**Metastasis**	↑ *VEGF, TRIO, SERPINB9, ITGA7, HGFR, TGFβ1, IL1B, MMP2, TWIST1, SNAIL1, CDH1, HIF-1α*	↓ *TIMP4, CDH1, CTNNB1, FN1*	[24,39,74,75,76,77,78,79,81]
**Malignant phenotype**	↑ *LCO4A1, ENO1, HK2, PGK1, MTFP1, HILPDA, VKORC1, TPI1, HIST1HIC*		[49]

Arrow pointing downwards: decrease in expression. Arrow pointing upwards: increase in expression.

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
