# Peer review of "Hypoxia in the Initiation and Progression of Neuroblastoma Tumours"

_ijms, 2019, doi:10.3390/ijms21010039_

Round 1

Reviewer 1 Report

The review “Hypoxia in the initiation and progression of neuroblastoma tumors” by Carlos Huertas-Castaño et al. describes the effects of hypoxia in NB progression and NB cell transformation and differentiation. The topic is of interest and the various subject matters discussed are rather comprehensive and relevant.

I have a couple of comments that should be addressed:

Since activation of the hypoxia response pathway may have relevant prognostic implications for survival in patients with NB, more information on studies that provide hypoxia signatures correlated with NB patients’ outcomes should be provided.

In addition, a table summarizing the hypoxia-mediated gene expression modulation associated with NB differentiation, cell survival, and tumor cell dissemination should be included.

Author Response

RESPONSE TO REVIEWER 1

We thank the reviewer for the comments. We already included information about hypoxia-related gene signature and survival in neuroblastoma (lines 221 onwards in the original manuscript). Following your suggestions, we have extended this section and include additional references on the issue (new references 50 and 51).

We have now included a Table (Table 1) with hypoxia-related gene expression changes and the processes in which they are involved. We hope this table helps summarize the findings.

Reviewer 2 Report

I find the review entitled “Hypoxia in the initiation and progression of neuroblastoma tumors” by Castano et al, is very interesting. It contains information on the study of hypoxia in neuroblastoma, a very hot topic that has not yet been fully investigated. This review has been divided into sessions, each concerning the role of hypoxia in different aspects of neuroblastoma. The information is very detailed. I would suggest to the authors to add more information about the regulation of gene expression in hypoxic neuroblastoma cells.

1. Hypoxia drives tumor plasticity through the acquisition of local or global chromatin modifications, which allow the accessibility of hypoxia-responsive elements (HRE) loci or of new active DNA regions at hypoxia inducible factors (HIFs) ( 2010;5:293–6). HIF1A is not the unique player to define the whole picture of hypoxia-regulated gene expression (Biochimie. 2017;135:28–34, Nature. 2016;537:63–8). Recent paper reports that neuroblastoma cells adapt to hypoxia by HIF1A-dependend and HIF1A-indipendent driven response. HIF1A driven transcriptional response in hypoxia is a consequence of the epigenetic control of low oxygen levels at DNA methylation status in intragenic and intergenic regions (BMC Med Genet.2019;20(1):37).

2. In “Hypoxia and NB cell survival” section, the authors should add details concerning HIF1-α and HIF-2α gene expression and neuroblastoma survival. Clinically, high levels of HIF1A and EPAS1 expression were associated with inferior survival in neuroblastoma, and patient subgroups with lower expression of HIF1A and EPAS1 showed significant enrichment of pathways related to neuronal differentiation, as described (Sci Rep.2015;5:11158 ). In this paper the authors suggest that the introduction of HIF inhibitors into differentiation therapy protocols might offer therapeutic advantages for relapse prevention. Indeed they show that  combination of all-trans retinoic acid (ATRA) with HIF1A or EPAS1 silencing led to an acquired glial-cell phenotype and enhanced expression of glial-cell differentiation markers and HIF1A or EPAS1 silencing might promote cell senescence independent of ATRA treatment.

3. In the line 216, the authors assert “potential hypoxia-independent functions of HIF-2α, related to aggressiveness, have also been suggested”. The authors should mention that proteomic analysis in neuroblastoma cells (J Proteome Res.2016;15:3643-3655) show that hypoxia-independent HIF-2α high expression correlates with an increment of fatty acid storage and a shift in global splicing regulation that has been previously reported in high-risk tumors (BMC Med Genomics. 2011; 4:35 ).

Author Response

RESPONSE TO REVIEWER 2

Thank you for the suggestion. We have now added the suggested information and references to our manuscript (line 277, new references 67-69).

We think this discussion would be more appropriately included in section 7 of the manuscript: “Hypoxia and related therapies”. We have commented on the suggested papers in the revised version of our manuscript (line 359 onwards and new reference 96).

We have added the suggested information and references on lines 215 onwards (new references 47 and 48).